

# LIMK1/2 inhibitor LIMKi 3 suppresses porcine oocyte maturation

Ru-Xia Jia, Xing Duan, Si-Jing Song and Shao-Chen Sun

College of Animal Science & Technology, Nanjing Agricultural University, Nanjing, China

## ABSTRACT

LIMKi 3 is a specific selective LIMK inhibitor against LIMK1 and LIMK2, while LIMK1 and LIMK2 are the main regulators of actin cytoskeleton to participate in many cell activities. However, the effect of LIMKi 3 in porcine oocyte meiosis is still unclear. The present study was designed to investigate the effects of LIMKi 3 and potential regulatory role of LIMK1/2 on porcine oocyte meiotic maturation. Immunofluorescent staining of p-LIMK1/2 antibody showed that LIMK1/2 was localized mainly to the cortex of porcine oocyte, which co-localized with actin. After LIMKi 3 treatment, the diffusion of COCs became weak and the rate of polar body extrusion was decreased. This could be rescued by moving oocytes to fresh medium. After prolonging the culture time of oocytes, the maturation rate of porcine oocyte increased in LIMKi 3 groups, indicating that LIMKi 3 may suppress the cell cycle during porcine oocyte maturation. We also found that after LIMKi 3 treatment actin distribution was significantly disturbed at porcine oocyte membranes and cytoplasm, indicating the conserved roles of LIMK1/2 on actin dynamics. Next we examined the meiotic spindle positioning in porcine oocyte, and the results showed that a majority of spindles were not attached to the cortex of porcine oocyte, indicating that LIMKi 3 may affect actin-mediated spindle positioning. Taken together, these results showed that LIMK1/2 inhibitor LIMKi 3 had a repressive role on porcine oocyte meiotic maturation.

# INTRODUCTION

The meiosis in mammalian oocyte is arrested at diplotene stage of first meiotic prophase, manifested by the germinal vesicle (GV) located at the center of the oocyte. Meiotic resumption from diplotene stage is characterized by germinal vesicle breakdown (GVBD). Chromatins condense into chromosomes and microtubules then are organized into the specialized barrel-shaped bipolar spindle. When all the chromosomes line up on the equatorial plate, oocyte enters metaphase I. Subsequently, the spindle relocates from the center of the oocyte to a position near the cortex in an actin filament-dependent manner (*Almonacid, Terret & Verlhac, 2014*). In addition, actin is enriched to form an actin cap over the spindle, and cortical granules (CGs) become redistributed to form a CG-free domain (CGFD) where microvilli are lost (*Deng et al., 2003*). And these events are regarded as cortical reorganization. After cortical reorganization, the contractile ring forms and then facilitates cytokinesis of the oocyte to generate a highly polarized egg with the emission of first polar body (*Liu, 2012*; *Maddox, Azoury & Dumont, 2012*).

Corresponding author
Shao-Chen Sun, sunsc@njau.edu.cn

Oocyte maturation is important for fertilization and following embryo development (*Maro & Verlhac, 2002*). Spindle and actin filaments are essential for the extrusion of polar body during the mammalian oocyte maturation (*Sun & Kim, 2013*).

LIM kinase (LIMK) which includes LIMK1 and LIMK2, belongs to serine kinases and is discovered more than a decade ago, and LIMK is shown to be a regulator of actin dynamics (*Stanyon & Bernard, 1999*). LIM kinase is distinguished by unique structural features: a protein kinase region at the C-terminal, a PDZ-like motif at the interposing region, and two tandemly arrayed LIM domains at the N-terminaland (*Ohashi et al., 1994*; *Okano et al., 1995*). Previous studies showed that LIMK regulated the actin polymerization mediated by the Rho family, including Rho (*Maekawa et al., 1999*), Rac (*Yang et al., 1998*), and Cdc42 (*Sumi et al., 1999*). Meanwhile, activated LIMK regulated actin reorganization via phosphorylation and inactivation of cofilin, a family of actin-binding proteins directly involved in the depolymerization and nucleation of actin filaments (F-actin) (*Arber et al., 1998*; *Bamburg, 1999*). The functions of LIMK contribute to its irreplaceable effects in cell movement, division and structure formation (*Bernard, 2007*). More extensively, LIMK actively promoted pathologic cancer cell division where drastic changes had taken place in actin filaments. HeLa cell as a model cancer cell has been used to investigate the roles of LIMK, and results showed that LIMK-mediated cofilin phosphorylation was required for precise spindle positioning to ensure the achievement of division during mitosis (*Kaji, Muramoto & Mizuno, 2008*). The regulation of LIMK is not only involved in mitosis but also meiosis. In Xenopus oocyte, XLIMK-cofilin system was required for the organization of the microtubule-derived precursor of the meiotic spindle (*Takahashi et al., 2001*). Additionally, LIMK activity was essential for microtubule organizing center organization and distribution in mouse oocyte meiosis (*Li et al., in press*). However, little is known regarding the study related with LIMK in porcine oocyte.

LIM kinase Inhibitor I, LIMKi 3, is a cell-permeable, potent inhibitor of LIM kinase (IC50 = 7 and 8 nM against LIMK1 and LIMK2, respectively) and shown to effectively destabilize F-actin structure in MDA-MB-231 breast cancer cells (3–10 µM) with concomitant blockage of invasion (by 93% at 10 µM). Here, we selected porcine as experiment animal to investigate the possible consequence of LIMKi 3 in oocyte meiosis. The treatment of LIMKi 3 significantly affected porcine oocyte maturation. We also showed that LIMKi 3 treatment caused aberrant actin distribution and spindle positioning, which might have contributed to a failure of porcine oocyte maturation.

## MATERIALS AND METHODS

### Antibodies and chemicals

Basic maturation culture medium was TCM 199 (Sigma, St. Louis, MO, USA). A rabbit polyclonal p-LIMK-1/2 antibody was from Santa Cruz (Santa Cruz, CA, USA). An Alexa Fluor 488 secondary antibody was from Invitrogen (Carlsbad, CA, USA). Phalloidin-TRITC and mouse monoclonal anti-$\alpha$-tubulin-FITC antibody were from Sigma. LIMK inhibitor LIMKi 3 was from Calbiochem (Darmstadt, Germany).

## Oocyte collection and culture

Animal use was conducted in accordance with the Animal Research Institute Committee guidelines of Nanjing Agricultural University, China. This study was specifically approved by the Committee of Animal Research Institute, Nanjing Agricultural University, China. Porcine ovaries were collected from a local slaughterhouse, and were then transported to laboratory within 3 h in sterile saline (0.9% NaCl) containing 500 IU/mL of penicillin and 500 IU/mL of streptomycin at 37 °C. Cumulus-oocyte complexes (COCs) were obtained from medium-sized follicles of ovaries by aspirating with a 20-gauge needle attached to a 5-ml disposable syringe. Oocytes surrounded by a compact cumulus mass and also with uniform ooplasm were selected and transferred to Dulbecco' s PBS, then washed three times with modified medium 199 containing 0.1% (wt/vol) polyvinyl alcohol, 0.91 mM sodium pyruvate, 3.05 mM glucose, 75 mg/L of penicillin, and 50 mg/L of streptomycin. Only oocytes with intact cumulus cells and evenly granulated ooplasm were chosen for *in vitro* maturation (IVM).

A group of 80 COCs were transferred to 500 µL of maturation medium consisting of 90% (vol/vol) modified M199, 10 ng/mL of EGF (mouse EGF; Sigma), 10 IU/mL of hCG, 10 IU/mL of pregnant mare's serum gonadotropin (PMSG), 0.57 mM cysteine (Sigma),and 10% (vol/vol) pig follicular fluid, and then covered with 200 µL paraffin oil in a four-well dish (NUNC) for 26 h (for COCs at metaphase I (MI)) or 44 h (for COCs at metaphase II (MII)) at 38.5 °C in a 5% $CO_2$ atmosphere.

## LIMK1/2 inhibitor LIMKi 3 treatment

For LIMK1/2 inhibitor treatment, stock LIMKi 3 (50 mM in dimethylsulfoxide (DMSO)) was diluted in M199 to final concentrations of 50, 100, 150 and 200 µM. A control group was cultured in DMSO at the same relative concentration of solvent. COCs were cultured with LIMKi 3 to evaluate its effects on oocyte maturation. COCs were denuded of their cumulus cells by gentle pipetting with 0.1% (w/v) hyaluronidase (Sigma). Oocytes with clearly extruded polar bodies were considered to be matured. The occurrence of first polar body extrusion in oocytes was examined after removing cumulus cells.

## Rescue experiment

After cultured in maturation medium M199 containing 200 µM LIMK inhibitor LIMKi 3 for 44 h, treated oocytes were washed three times in fresh culture solution (2 min each wash). Oocytes were then transferred to fresh culture solution and cultured for an additional 16 h under paraffin oil at 38.5 °C in a 5% $CO_2$ atmosphere.

## Immunofluoorescence staining and confocal microscopy

For immunofluorescence staining of LIMK1/2, actin, and $\alpha$-tubulin, oocytes were fixed in 4% paraformaldehyde in PBS at room temperature for 30 min and then transferred to a membrane permeabilization solution (0.5% Triton X-100) for 8–12 h at room temperature. To suppress nonspecific binding of IgG, oocytes were blocked in blocking buffer for 1 h at room temperature (1% BSA-supplemented PBS). Then oocytes were incubated overnight at 4 °C or 5 h at room temperature with a rabbit polyclonal p-LIMK-1/2 antibody (1:50) or anti-$\alpha$-tubulin-FITC (1:200) for 3 h or with 2 µg/mL of Phalloidin-TRITC at room

temperature for 1 h. After three washes (2 min each) in wash buffer (0.1% Tween 20 and 0.01% Triton X-100 in PBS), oocytes were labeled with Alexa Fluor-488 goat anti-rabbit IgG (1:100; for p-LIMK-1/2 staining) for 1 h at room temperature. Oocytes were stained with Hoechst 33342 for 10 min, mounted on glass slides, and then examined with a confocal laser-scanning microscope (Zeiss LSM 700 META; Zeiss, Oberkochen, Germany). Each experiment was repeated three times; at least 30 oocytes were examined per experimental condition.

## Fluorescence intensity analysis

To analyze the fluorescence intensity of actin filaments, samples of control and treated oocytes were mounted on the same glass slide. Image J software was used to define a region of interest (ROI), and the average fluorescence intensity per unit area within the ROI was determined. Independent measurements using identically sized ROIs were taken for the cell membrane and cytoplasm. The average values of all measurements were used to determine the final average intensity between control and treated oocytes.

## Western blot analysis

A total of 100 porcine oocytes were collected, lysed in Laemmli sample buffer (SDS sample buffer with 2-mercaptoethanol), and boiled at 100 °C for 10 min. Proteins were separated by SDS-PAGE and then electrophoretically transferred to polyvinylidene fluoride membranes. To avoid nonspecific binding, membranes were blocked with Tris-buffered saline (TBS) containing 0.1% (w/w) Tween 20 (TBST) and 5% (w/v) nonfat dry milk powder for 1 h at room temperature. The membranes were simultaneously incubated at overnight 4 °C with a rabbit polyclonal p-LIMK-1/2 (1:200, Santa Cruz, USA) or rabbit monoclonal anti-$\alpha$-tubulin antibody (1:2,000; Cell Signaling Technology, Danvers, MA, USA). After washing three times in TBST (10 min each), the membranes were incubated for 2 h with secondary anti-rabbit HRP-conjugated antibodies (1:2,000; Cell Signaling Technology, Beverly, MA, USA) in 5% nonfat dry milk in TBST at room temperate. Finally, the membranes were washed 3 times in TBST and then the specific proteins were visualized using chemiluminescence reagent (Millipore, Billerica, MA).

## Statistical analysis

A group of 80 oocytes were cultured with or without LIMKi 3 for once experimental condition and at least three replicates were performed for each treatment. Results were expressed as means $\pm$ standard errors of the mean. Statistical analysis was conducted using analysis of variance (ANOVA), and differences between treatment groups were assessed by Duncan's multiple comparison test. $P < 0.05$ was considered significant.

# RESULTS

## P-LIMK1/2 expression in porcine oocyte

The subcellular localizations of p-LIMK-1/2 at different stages of meiosis were assessed by immunofluorescent staining. As shown in Fig. 1A, p-LIMK-1/2 was primarily localized at cortex of oocyte during all meiotic stages, and this localization was similar with actin.

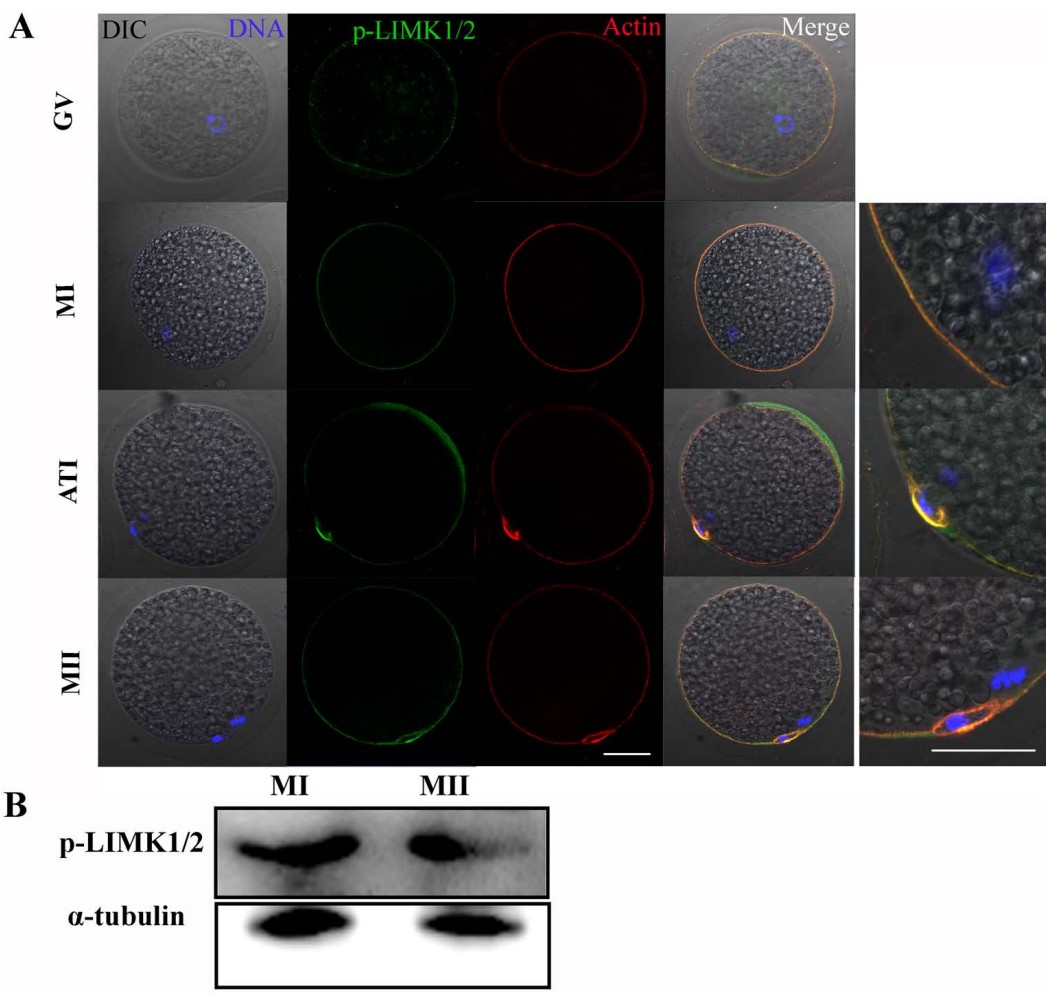

**Figure 1** **p-LIMK-1/2 expression in porcine oocytes.** (A) Subcellular localization of p-LIMK-1/2 during porcine oocyte meiotic maturation. Double staining of p-LIMK-1/2 and actin, we found that P-LIMK-1/2 accumulated at the cortex of oocytes from GV to the MII stage, which was co-localized with actin. Green, p-LIMK-1/2; red, actin; blue, chromatin. Bar = 30 µm. (B) The protein expression of p-LIMK-1/2 in porcine oocyte. Using western blot, the result showed that p-LIMK-1/2 was all expressed at the MI and MII stages.

Oocytes of GV stage were obtained and were cultured for 26 and 44 h, which corresponded to the times to achieve the MI and MII stages, respectively. Using western blot, we found that p-LIMK-1/2 protein was all expressed at the MI and MII stages (Fig. 1B).

## LIMKi 3 treatment suppresses polar body extrusion *in vitro*

Based on the examination of LIMK1/2 expression and localization in porcine oocytes, we next examined whether LIMK1/2 inhibitor LIMKi 3 affected porcine oocyte maturation. As shown in Fig. 2A, COCs exhibited different diffusion situation between control and treatment group: the diffusion of 200 µM LIMKi 3 treated-COCs was significantly weaker than the control group. In the standard of COCs surrounded by the cumulus cell layers, we counted the proportion of diffusion and found that the rate of good diffusion was decreased

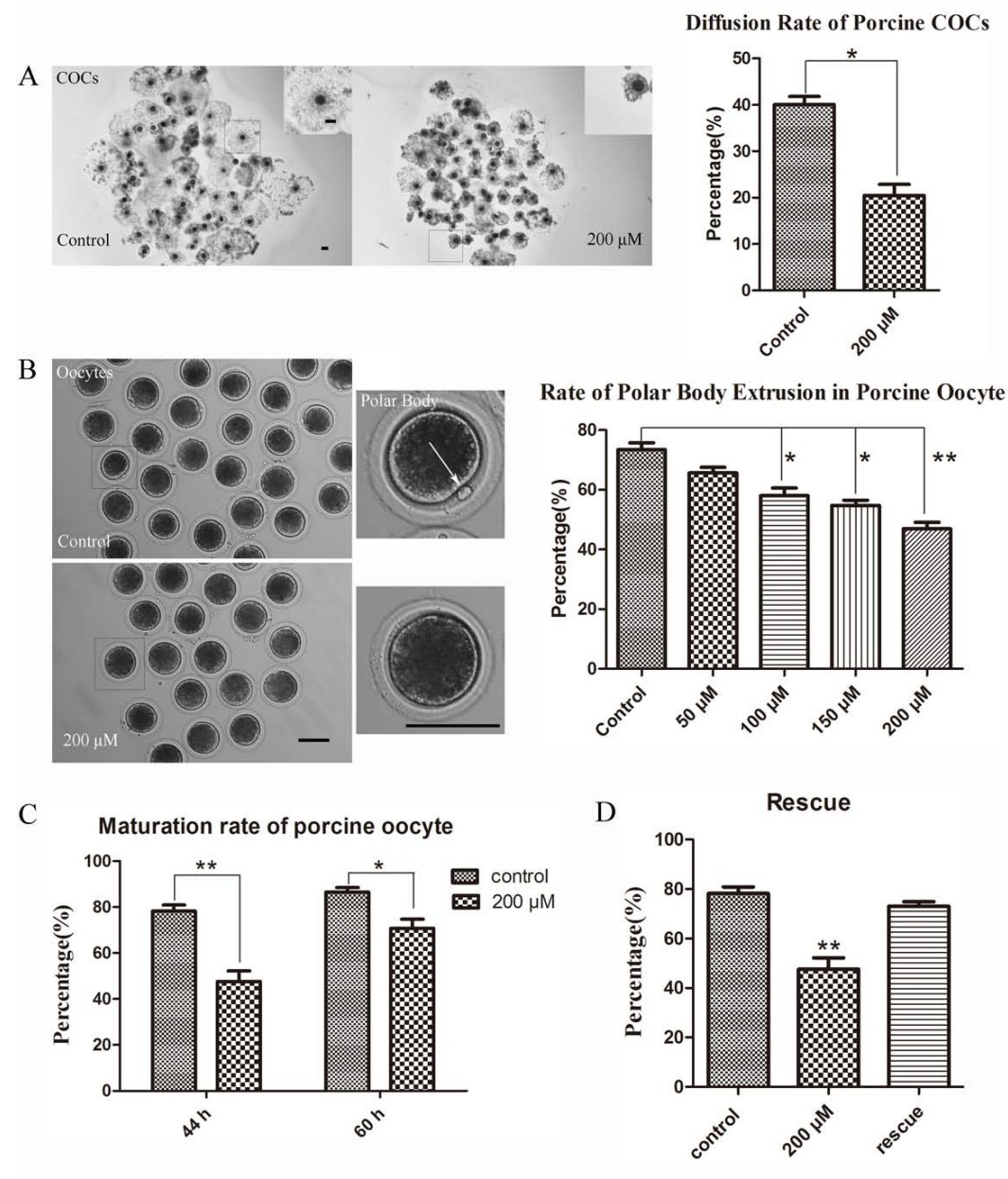

**Figure 2 LIMKi 3 treatment suppresses porcine oocyte maturation *in vitro* (A) LIMKi 3 treatment efficiently restrained the diffusion of COCs.** For COCs, cumulus cell expansion was weak after treatment with LIMKi 3; with significantly decreased diffusion rate of COCs ($p < 0.05$). Bar = 100 µm (B) LIMKi 3 treatment affected the polar body excrusion of oocyte in porcine. Most oocytes of control were able to extrude polar body, whereas few oocytes extruded polar body after LIMKi 3 treatment and the rate of polar body extrusion decreased in a dose-dependent manner. Contrast to control group, the ratio of polar body extrusion was significantly lower with LIMKi 3 treatment at 100 µM ($p < 0.05$), 150 µM ($p < 0.05$) and 200 µM ($P < 0.01$). Bar = 100 µm (C) Compared with treated oocytes in 44 h, after prolonging the culture time the maturation rate increased ($70.68 \pm 6.99\%$ vs. $47.61 \pm 7.92\%$). (D) After rescuing oocytes from LIMKi 3, the maturation rate of porcine oocyte increased.

significantly ($40.05 \pm 3.03\%$ vs. $20.50 \pm 4.05\%$; $p < 0.05$). Oocytes were then obtained from COCs after hyaluronidase treatment, we found that few oocytes extruded the polar body after 200 μM LIMKi 3 treatment. In addition, the rate of polar body extrusion of oocytes was reduced in a dose-dependent manner (Fig. 2B). For controls, the ratio of polar body extrusion was $73.44 \pm 5.18\%$ ($n = 123$); while oocytes were treated with LIMKi 3 in the concentration of 50 μM, 100 μM, 150 μM, 200 μM, the ratio decreased to $65.66 \pm 3.24\%$ ($n = 95$; $p > 0.05$), $58.04 \pm 5.58\%$ ($n = 111$; $p < 0.05$), $54.76 \pm 3.00\%$ ($n = 101$; $p < 0.05$), $46.88 \pm 5.04\%$ ($n = 135$; $p < 0.01$). Therefore, the porcine oocyte maturation was suppressed in the treatment of LIMKi 3, and 200 μM was chosen as the appropriate concentration for subsequent experiments.

To further clarify the effects of LIMKi 3 on porcine oocyte maturation, we prolonged the culture time of porcine oocyte to 60 h. As shown in Fig. 2C, after prolonging the culture time, the maturation rate increased compared with 44 h culture group ($70.68 \pm 6.99\%$ vs. $47.61 \pm 7.92\%$). Thus, LIMKi 3 repressed the cell cycle of porcine oocyte maturation. To further confirm the inhibitory effects of LIMKi 3, we did the rescue experiment. Compared with oocytes before rescue ($78.18 \pm 4.69\%$ vs. $47.61 \pm 7.92\%$; $p < 0.01$), LIMKi 3 treated oocytes reached the MII stage (200 μM) after an additional 16 h of culture in fresh medium, with the ratio of $73.06 \pm 3.09\%$ (Fig. 2D). Therefore, LIMKi 3 may have a repressive effect on porcine oocyte maturation and its effect was reversible.

## LIMKi 3 treatment causes disruption of actin distribution in porcine oocyte

Considering that LIMK1/2 has been known to regulate actin cytoskeleton, we next examined the effects of LIMK inhibitor LIMKi 3 on actin filament distribution of meiosis I in porcine oocyte. As shown in Fig. 3A (10X), compared with the control oocytes, actin fluorescence intensity of LIMKi 3 treated oocytes became much weaker. Furthermore, actin distribution was disturbed both at the membrane and cytoplasm of oocyte after LIMKi 3 treatment (Fig. 3A (40X)). The value of actin fluorescence intensity in porcine oocyte was analyzed by Image J software (Fig. 3B). For membrane actin fluorescence intensity, the rate of LIMKi 3 treated oocytes was significantly lower than that in control oocytes ($78.88 \pm 16.53\%$ vs. $48.18 \pm 9.14\%$; $P < 0.01$). Compared with control oocytes, the rate of cytoplasmic actin also decreased after LIMKi 3 treatment ($29.76 \pm 9.02\%$ vs. $22.70 \pm 7.81\%$; $P < 0.01$). These results indicated that the LIMKi 3 treatment resulted in an abnormal actin distribution.

## LIMKi 3 treatment causes abnormity of spindle positioning in porcine oocyte

Spindle migration in meiosis I is actin dependent, so we next investigated the effects of LIMK1/2 inhibitor LIMKi 3 on spindle positioning at MI stage. As shown in Fig. 4A (10X), most of the spindles exhibited at the position distant with the cortex after treatment with 200 μM LIMKi 3, which is different with the spindles of control oocytes that exhibited peripheral localization. Enlarged images more clearly showed the difference between control and treatment (Fig. 4A (40X)).

We also analyzed the cell cycle stage after culture for 26 h. As shown in Fig. 4B, the percentage of oocytes with spindles closed to cytoplasm center at the MI stage for LIMKi 3

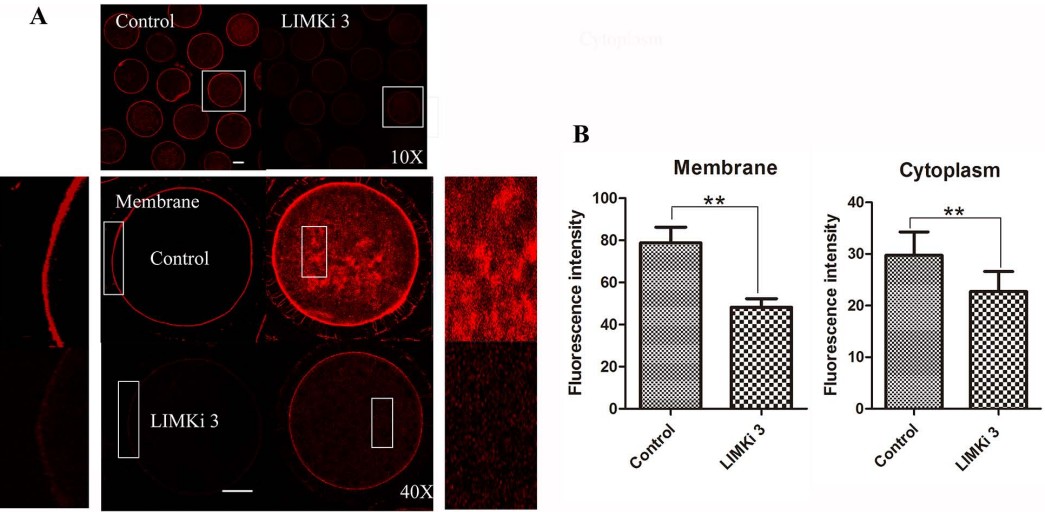

**Figure 3** **LIMKi 3 treatment causes disruption of actin distribution in porcine oocyte.** (A) Actin distribution of MI oocytes was examined by immunofluorescent staining. Oocytes that were treated with 200 μM LIMKi 3 had profoundly disterbed actin distribution. Both cortical actin and cytoplasmic actin were not visible in treated oocytes. Bar = 30 μm (B) Average actin fluorescence intensity of MI oocytes. Actin fluorescence intensity was significantly lower in LIMKi 3-treated oocytes at the membrane ($P < 0.01$) and in the cytoplasm ($P < 0.01$).

treatment was clearly higher than that of control oocytes ($57.11 \pm 7.85\%$, $n = 212$ vs. $26.58 \pm 5.18\%$, n=234) ($p < 0.01$), nevertheless the percentage of oocytes with spindles at the cortex in the MI stage ($35.27 \pm 7.94\%$ vs. $59.16 \pm 11.19\%$; $p < 0.01$) and in the anaphase/telophase (AT) stage ($3.27 \pm 4.53\%$ vs. $9.61 \pm 6.04\%$; $p < 0.05$) were reduced after LIMKi 3 treatment. Meanwhile, the percentage of oocytes at the GVBD stage showed no significant difference between control and treatment ($4.65 \pm 3.08\%$ vs. $4.35 \pm 4.00\%$; $p > 0.05$). These results demonstrated that LIMKi 3 treatment effectively affected spindle positioning.

## DISCUSSION

In present study, we investigated the potential roles of LIMK1/2 inhibitor LIMKi 3 during porcine oocyte maturation. The results showed that LIMKi 3 treatment inhibited the diffusion of COCs and the emission of polar body, which affected porcine oocyte maturation. Moreover, the inhibition of LIMKi 3 disrupted actin distribution and meiotic spindle positioning. We speculated that LIMK1/2 inhibitor LIMKi 3 suppressed porcine oocyte maturation through its effects on actin-based spindle positioning.

We first examined LIMK1/2 expression in porcine oocyte, and we found that LIMK1/2 existed at the cortex of oocyte in porcine during meiosis, which was similar with actin filament. We next examined whether LIMK1/2 inhibitor LIMKi 3 affected porcine oocyte maturation. After LIMKi 3 treatment, the diffusion of COCs became weak, together with the decreased rate of polar body extrusion, these results indicated that LIMKi 3 suppressed porcine oocyte maturation. In COCs, there is a mutual promoting relationship between oocyte and granulosa cells. Oocyte is crucial in the granulosa differentiation

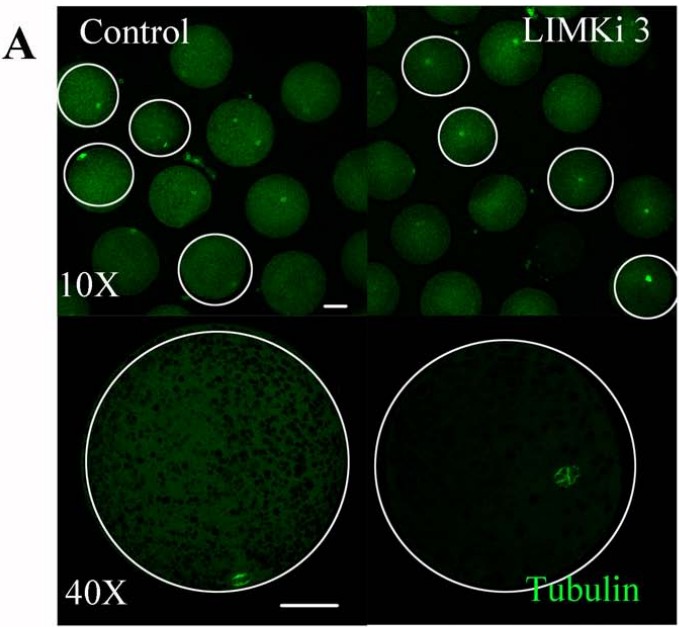

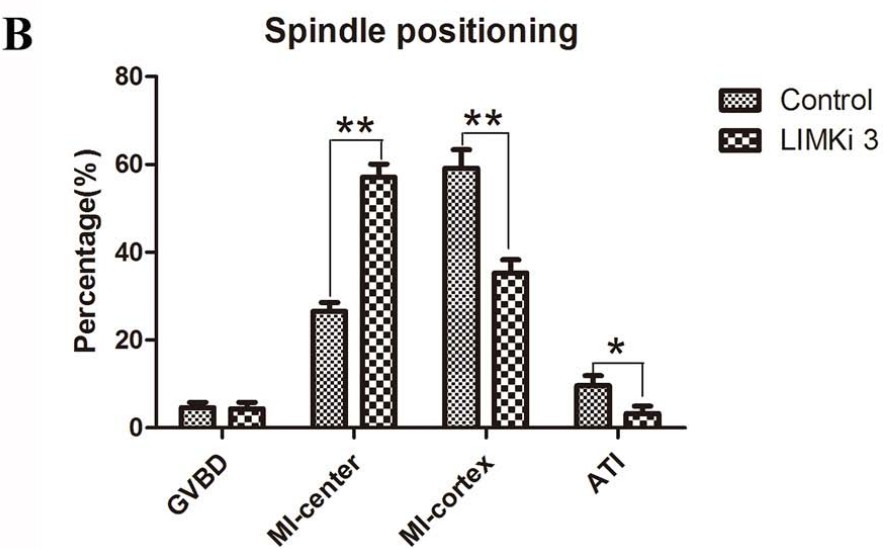

**Figure 4** **LIMKi 3 treatment causes abnormity of spindle position in porcine oocyte.** (A) Spindle localization in oocyte. For controls, spindles were located peripherally, whereas they were nearly centrally located in LIMKi 3 treated oocytes. Enlarged images below show the distances of spindles from the cortex. Bar = 30 μm (B) Spindle stages and positioning after 26 h in culture. Contrast with controls, the number of LIMKi 3 treated-oocytes with peripherally located spindles was significant lower ($p < 0.01$).

and development (*Gilchrist, Lane & Thompson, 2008*). Likewise, granulosa cells also play regulating roles in the nutrition and signal transduction for oocyte development and maturation (*Sutton-McDowall, Gilchrist & Thompson, 2010*). Therefore, it is hard to examine whether the suppression of oocyte maturation by LIMKi 3 is through the effects of LIMKi 3 on cumulus expansion. Similarly, LIMK has been found to involve oocyte

maturation in Xenopus (*Takahashi et al., 2001*). Upon microinjection of Xenopus-Limk (Xlimk1/2), abundant LIMK significantly inhibited the appearance of a white maturation spot (WMS), an indicator of entry into Xenopus oocyte meiosis, which indicated the failure of Xenopus oocyte maturation. Additionally, the activators of LIMK, which include Rho (*Zhang et al., 2014a*), Rac (*Halet & Carroll, 2007*) and Cdc42 (*Dehapiot et al., 2013*; *Ma et al., 2006*), are also required for polar body extrusion during the meiotic maturation of mammalian oocyte. Therefore, our results showed that LIMK1/2 inhibitor LIMKi 3 could suppress the polar body extrusion during porcine oocyte maturation.

LIMK catalyzes phosphorylation of an N-terminal 3rd serine residue of cofilin and inhibits its activity to depolymerize actin, thereby leading to stabilization of actin (*Arber et al., 1998*; *Yang et al., 1998*). Our results showed similar localization pattern of LIMK1/2 and actin filaments. We then investigated whether LIMK1/2 inhibitor LIMKi 3 participated in the regulation of actin filament during porcine oocyte meiotic maturation. Staining with phalloidin-TRITC we found that LIMKi 3 treatment affected actin distribution in the membrane and cytoplasm of porcine oocyte showing with the decreased rate of actin fluorescence intensity. It has been reported that actin flow droves oocytes progressed to MII stage, which is characterized by asymmetric polar body emission of oocyte (*Yi & Li, 2012*). Thus, the results demonstrated that LIMK also regulated the actin distribution, which further disrupted polar body extrusion in porcine oocyte meiosis. Similar results were observed with Rho-kinase (ROCK), the upstream protein of LIMK1/2, which was shown to be responsible for actin filament distribution which contributed to asymmetric division in meiotic oocyte (*Duan et al., 2014*; *Zhang et al., 2014b*). Therefore, our results indicated that the effects of LIMK1/2 inhibitor LIMKi 3 to porcine oocyte maturation may be triggered by its effects on actin distribution.

Actin drives meiotic spindle migration and anchor to ensure spindle positioning accurately, which is required for the achievement of oocyte nuclear maturation (*Almonacid, Terret & Verlhac, 2014*; *Brunet & Verlhac, 2011*). Here, we next examined the positioning of spindle after LIMKi 3 treatment. The results showed that most spindles of inhibitor-treated oocytes primarily localized at the position distant with the cortex after 26 h in culture, the time point by which the meiotic spindles of most oocytes should have attached to the cortex or even entered anaphase I. The results indicated the failure of spindle movement after LIMKi 3 treatment during porcine oocyte meiotic maturation. Our results were similar with the study on bovine which showed the ROCK/LIMK/cofilin pathway to meiotic spindle migration during oocyte maturation. Therefore, LIMK1/2 inhibitor LIMKi 3 may affect spindle positioning via regulation of actin dynamic during oocyte meiotic maturation in porcine.

In conclusion, our results demonstrated that LIMK1/2 inhibitor LIMKi 3 treatment disrupted actin microfilaments distribution and caused the failure of spindle positioning, which affected porcine oocyte maturation.

### Funding

This study was supported by the National Basic Research Program of China (2014CB138503); the National Natural Science Foundation of China (31571547, 31622055); the Natural Science Foundation of Jiangsu Province (BK20140030), China; and the Fundamental Research Funds for the Central Universities (KJYQ201701, KJJQ201501, KJTZ201602). The funders had no role in study design, data collection and analysis, decision to publish, or preparation of the manuscript.

### Grant Disclosures

The following grant information was disclosed by the authors:
The National Basic Research Program of China: 2014CB138503.
The National Natural Science Foundation of China: 31571547, 31622055.
The Natural Science Foundation of Jiangsu Province: BK20140030.
The Fundamental Research Funds for the Central Universities: KJYQ201701, KJJQ201501, KJTZ201602.

### Competing Interests

Shao-Chen Sun is an Academic Editor for PeerJ.

### Author Contributions

- Ru-Xia Jia conceived and designed the experiments, performed the experiments, analyzed the data, wrote the paper, prepared figures and/or tables.
- Xing Duan analyzed the data, contributed reagents/materials/analysis tools.
- Si-Jing Song performed the experiments, contributed reagents/materials/analysis tools.
- Shao-Chen Sun conceived and designed the experiments, analyzed the data, reviewed drafts of the paper.

### Animal Ethics

The following information was supplied relating to ethical approvals (i.e., approving body and any reference numbers):

Animal use was conducted in accordance with the Animal Research Institute Committee guidelines of Nanjing Agricultural University, China (Jiangsu 2008-45). This study was specifically approved by the Committee of Animal Research Institute, Nanjing Agricultural University, China.

### Data Availability

The raw data has been supplied as Data S1.

### Supplemental Information

Supplemental information for this article can be found online at http://dx.doi.org/10.7717/peerj.2553#supplemental-information.

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
