# Peer review of "LIMK1/2 inhibitor LIMKi 3 suppresses porcine oocyte maturation"

_PeerJ, doi:10.7717/peerj.2553_

## Round 0.1 · original submission · Major Revisions

This study is potentially interesting, but additional experiments are required in order to confirm somewhat preliminary results. These should address namely the effect of the inhibitor on progression of meiosis I and on cumulus expansion, as well as the reversibility of the inhibitor, as requested by both reviewers. The manuscript requires also extensive language editing.

Reviewer 1 ·

Basic reporting

The MS is written clearly. English needs some improvement. The aim of the study is clearly stated and its rationale is given in Introduction. Relevant literature is cited. Study is very simple so it is easy to understand. MS conforms to PeerJ standard. Figures are relevant and appropriately labeled and described. Raw data are supplied.

Experimental design

Research performed by authors is original however it is very limited and superficial. Very limited set of experiments was performed and some doubts and many unanswered questions remain.
It is stated, e.g. in in the legend of Figure 2, that oocyte maturation is efficiently blocked by LIMK1/2. Is it sure? There could be only delay in oocyte maturation and maturation could be completed when culture is prolonged beyond 44 hours. This suspicion is supported by only limited portion of oocytes prevented from reaching MII. Maturation seems rather to be suppressed.
Only one inhibitor was used to demonstrate the effects of LIMK1/2 on porcine oocyte maturation. Experiments with at least one more structurally different inhibitor will give more confidence to this study. Another ways for blocking or activation of LIMK1/2 should be considered. As usual in inhibitors, LIMKi3 has its side effects. LIMKi3 is e.g. interacting with microtubule (Ross-Mcdonald et al., 2008) and this effect could be involved in observed effect attributed by authors to the effect of LIMKi3 on LIMK1/2 and/or on actin filaments. What are the effects of LIMKi3 on oocyte maturation when actin filaments are stabilized by some actin-stabilizing agents? Demonstration of changes in cofilin phosphorylation will be useful.
There is stated in Results that “with 200 M LIMKi3 the expansion of COCs was weaker” and in Discussion it is stated that “expansion of COCs … failed” under the effect of inhibitor. However, these statements are not supported by any data, e.g. measurement of area of expanded COCs or measurement of hyaluronic acid produced during cumulus expansion. Data about cumulus expansion are needed and significant difference has to be observed for statement that LIMK1/2 is involved in cumulus expansion. Figure 2 indicate that cumulus expansion could be altered by LIMKi3-treatment however there are still visible COCs with marked cumulus expansion.
In Figure 1 GV oocytes are shown with localization of DNA, LIMK1/2 and actin. However, WBs are presented only for MI and MII oocytes and WB of GV oocytes is missing. Why? It should be more informative to present WB for all three stages investigated including GV.
Figure 2. It seems that there are two bands for actin in MII oocytes? Why?

Validity of the findings

Validity of findings is compromised by limited set of experiments. More convincing data has to be given to support really specific effects of LIMKi3 and to support the validity of conclusions on LIMK1/2 role in porcine oocyte maturation.
Since LIMKi3 only lowered ratio of oocytes maturing to MII and about half oocytes matured under the effect of 200 M of inhibitor, the declaration “LIMK1/2 is essential for porcine meiotic maturation” sounds too strong. LIMK1/2 is probably involved, but does not seem to be “essential”.
It is not clear why results of this study are discussed with studies investigating role of formin-2, Spire and other factors and why authors conclude their “results show accordance with these studies”. In the best case, this part of Discussion could be used for the suggestions for further research. It does not support findings described in this study.
There are too many speculations based on very limited set of data.

Additional comments

No comments

Reviewer 2 ·

Basic reporting

There are multiple English grammatical errors and typos that need to be corrected in the current manuscript. That authors can improve the quality of the manuscript by having someone edit it. These include, but are not limited to the following sentences: Lines 52, 58, 67, 224-226, etc.

The figures require better labeling and more details. For example, the name of the LIMK inhibitor should be stated in each figure.

In the abstract what does "indicating the conserved roles of LIMK1/2 on actin dynamics between species" mean? Only one species was evaluated in the manuscript.

Experimental design

The authors must provide additional dose data on the LIMK inhibitor. Only two higher concentrations are used. The IC50 of this compound is significantly lower and in cell culture lower doses are typically reported in the literature. The authors can diminish this concern by provide a range of doses. Is treatment with the inhibitor reversible? Authors should consider providing data on an experiment where the drug is removed part way through the experiment (wash out).

Validity of the findings

For the most part the experiments were performed well with sufficient numbers of oocytes. Additional experiments described above will strengthen the findings.

The authors should elaborate on the effects of LIMK inhibitor on cumulus expansion. Is this an indirect effect or a direct effect? Oocytes can still mature in the absence of cumulus cells. This should be discussed under the discussion.

---

## Round 0.2 · Major Revisions

Although this version of manuscript was improved in many aspects, including the analysis of cumulus expansion and dose-dependent impact on PBE, the effect of LIMKi 3 is still unclear and the possibility of meiosis I deceleration cannot be excluded. Especially considering the low effect on PBE rate - even at 200 μM 47% of cells still underwent PBE. Additional experiments are therefore required addressing the PBE in intervals after 44 hours and optionally also the reversibility of the inhibitor.

Reviewer 1 ·

Basic reporting

Authors revised their MS. They have improved text and have made appropriate corrections. With the exception of objective assessment of cumulus expansion they reject all suggestions on experimental design. Data remained only preliminary and unconvincing.

Experimental design

Experimental design does not allow to conclude on the role of LIMK1/2 on pig oocyte maturation.

Validity of the findings

Findings are only preliminary.

Additional comments

I suggest to reject MS.

Reviewer 2 ·

Basic reporting

The quality of the manuscript was greatly improved following the revision.

Experimental design

The authors improved the manuscript by including additional data.

Validity of the findings

The conclusions are now supported by the data. Additional sentences were added to the discussion.

---

## Round 0.3 · accepted · Accept

This version of the manuscript was significantly improved by additional experiments requested by both reviewers and changes to the manuscript implemented by authors and I feel that the message of the manuscript is now firmly supported by experimental data

Reviewer 2 ·

Basic reporting

The manuscript meets the standards.

Experimental design

Robust experimental design with proper controls.

Validity of the findings

The authors now provide a crucial rescue experiment that supports the findings in the manuscript.These experiments were originally proposed by reviewer 1 and 2.